# Online Learning of Nonparametric Mixture Models via Sequential Variational Approximation

**Dahua Lin**
Toyota Technological Institute at Chicago
`dhlin@ttic.edu`

## Abstract

Reliance on computationally expensive algorithms for inference has been limiting the use of Bayesian nonparametric models in large scale applications. To tackle this problem, we propose a Bayesian learning algorithm for DP mixture models. Instead of following the conventional paradigm – random initialization plus iterative update, we take an progressive approach. Starting with a given prior, our method recursively transforms it into an approximate posterior through sequential variational approximation. In this process, new components will be incorporated on the fly when needed. The algorithm can reliably estimate a DP mixture model in one pass, making it particularly suited for applications with massive data. Experiments on both synthetic data and real datasets demonstrate remarkable improvement on efficiency – orders of magnitude speed-up compared to the state-of-the-art.

## 1   Introduction

Bayesian nonparametric mixture models [7] provide an important framework to describe complex data. In this family of models, *Dirichlet process mixture models (DPMM)* [1, 15, 18] are among the most popular in practice. As opposed to traditional parametric models, DPMM allows the number of components to vary during inference, thus providing great flexibility for explorative analysis. Nonetheless, the use of DPMM in practical applications, especially those with massive data, has been limited due to high computational cost. MCMC sampling [12, 14] is the conventional approach to Bayesian nonparametric estimation. With heavy reliance on local updates to explore the solution space, they often show slow mixing, especially on large datasets. Whereas the use of split-merge moves and data-driven proposals [9, 17, 20] has substantially improved the mixing performance, MCMC methods still require many passes over a dataset to reach the equilibrium distribution.

Variational inference [4, 11, 19, 22], an alternative approach based on mean field approximation, has become increasingly popular recently due to better run-time performance. Typical variational methods for nonparametric mixture models rely on a truncated approximation of the stick breaking construction [16], which requires a fixed number of components to be maintained and iteratively updated during inference. The truncation level are usually set conservatively to ensure approximation accuracy, incurring considerable amount of unnecessary computation.

The era of *Big Data* presents new challenges for machine learning research. Many real world applications involve massive amount of data that even cannot be accommodated entirely in the memory. Both MCMC sampling and variational inference maintain the entire configuration and perform iterative updates of multiple passes, which are often too expensive for large scale applications. This challenge motivated us to develop a new learning method for Bayesian nonparametric models that can handle massive data efficiently. In this paper, we propose an online Bayesian learning algorithm for generic DP mixture models. This algorithm does not require random initialization of components. Instead, it begins with the prior $\mathrm{DP}(\alpha\mu)$ and progressively transforms it into an approximate posterior of the mixtures, with new components introduced on the fly as needed. Based on a new way of variational approximation, the algorithm proceeds sequentially, taking in one sample at a time to make the

update. We also devise specific steps to prune redundant components and merge similar ones, thus further improving the performance. We tested the proposed method on synthetic data as well as two real applications: modeling image patches and clustering documents. Results show empirically that the proposed algorithm can reliably estimate a DP mixture model in a single pass over large datasets.

## 2 Related Work

Recent years witness lots of efforts devoted to developing efficient learning algorithms for Bayesian nonparametric models. A n important line of research is to accelerate the mixing in MCMC through better proposals. Jain and Neal [17] proposed to use split-merge moves to avoid being trapped in local modes. Dahl [6] developed the sequentially allocated sampler, where splits are proposed by sequentially allocating observations to one of two split components through sequential importance sampling. This method was recently extended for HDP [20] and BP-HMM [9].

There has also been substantial advancement in variational inference. A significant development along is line is the *Stochastic Variational Inference*, a framework that incorporates stochastic optimization with variational inference [8]. Wang *et al.* [23] extended this framework to the non-parametric realm, and developed an online learning algorithm for HDP [18]. Wang and Blei [21] also proposed a truncation-free variational inference method for generic BNP models, where a sampling step is used for updating atom assignment that allows new atoms to be created on the fly.

Bryant and Sudderth [5] recently developed an online variational inference algorithm for HDP, using mini-batch to handle streaming data and split-merge moves to adapt truncation levels. They tried to tackle the problem of online BNP learning as we do, but via a different approach. First, we propose a generic method while they focuses on topic models. The designs are also different – our method starts from scratch and progressively adds new components. Its overall complexity is $O(nK)$, where $n$ and $K$ are number of samples and expected number of components. Bryant's method begins with random initialization and relies on splits over mini-batch to create new topics, resulting in the complexity of $O(nKT)$, where $T$ is the number of iterations for each mini-batch. The differences stem from the theoretical basis – our method uses sequential approximation based on the predictive law, while theirs is an extension of the standard truncation-based model.

Nott *et al.* [13] recently proposed a method, called VSUGS, for fast estimation of DP mixture models. Similar to our algorithm, the VSUGS method proposed takes a sequential updating approach, but relies on a different approximation. Particularly, what we approximate is a joint posterior over both data allocation and model parameters, while VSUGS is based on the approximating the posterior of data allocation. Also, VSUGS requires fixing a truncation level $T$ in advance, which may lead to difficulties in practice (especially for large data). Our algorithm provides a way to tackle this, and no longer requires fixed truncation.

## 3 Nonparametric Mixture Models

This section provide a brief review of Dirichlet Process Mixture Model – one of the most widely used nonparametric mixture models. A *Dirichlet Process (DP)*, typically denoted by $\mathrm{DP}(\alpha\mu)$ is characterized by a *concentration parameter* $\alpha$ and a *base distribution* $\mu$. It has been shown that sample paths of a DP are almost surely discrete [16], and can be expressed as

$$D = \sum_{k=1}^{\infty} \pi_k \delta_{\phi_k}, \quad \text{with } \pi_k = v_k \prod_{l=1}^{k-1} v_l, \ v_k \sim \text{Beta}(1, \alpha_k), \ \forall k = 1, 2, \ldots. \tag{1}$$

This is often referred to as the *stick breaking representation*, and $\phi_k$ is called an *atom*. Since an atom can be repeatedly generated from $D$ with positive probability, the number of distinct atoms is usually less than the number of samples. The *Dirichlet Process Mixture Model (DPMM)* exploits this property, and uses a DP sample as the prior of component parameters. Below is a formal definition:

$$D \sim DP(\alpha\mu), \quad \theta_i \sim \mu, \ x_i \sim F(\cdot|\theta_i), \ \forall i = 1, \ldots, n. \tag{2}$$

Consider a partition $\{C_1, \ldots, C_K\}$ of $\{1, \ldots, n\}$ such that $\theta_i$ are identical for all $i \in C_k$, which we denote by $\phi_k$. Instead of maintaining $\theta_i$ explicitly, we introduce an indicator $z_i$ for each $i$ with

$\theta_i = \phi_{z_i}$. Using this clustering notation, this formulation can be rewritten *equivalently* as follows:

$$z_{1:n} \sim \mathrm{CRP}(\alpha), \quad \phi_k \sim \mu, \quad \forall k = 1, 2, \ldots K$$
$$x_i \sim F(\cdot | \phi_{z_i}), \quad \forall i = 1, 2, \ldots, n. \tag{3}$$

Here, $\mathrm{CRP}(\alpha)$ denotes a *Chinese Restaurant Prior*, which is a distribution over exchangeable partitions. Its probability mass function is given by

$$p_{CRP}(z_{1:n}|\alpha) = \frac{\Gamma(\alpha)\alpha^K}{\Gamma(\alpha+n)} \prod_{k=1}^{K} \Gamma(|C_k|). \tag{4}$$

# 4  Variational Approximation of Posterior

Generally, there are two approaches to learning a mixture model from observed data, namely *Maximum likelihood estimation (MLE)* and *Bayesian learning*. Specifically, *maximum likelihood estimation* seeks an optimal point estimate of $\nu$, while *Bayesian learning* aims to derive the posterior distribution over the mixtures. Bayesian learning takes into account the uncertainty about $\nu$, often resulting in better generalization performance than MLE.

In this paper, we focus on Bayesian learning. In particular, for DPMM, the predictive distribution of component parameters, conditioned on a set of observed samples $x_{1:n}$, is given by

$$p(\theta'|x_{1:n}) = \mathbb{E}_{D|x_{1:n}} \left[ p(\theta'|D) \right]. \tag{5}$$

Here, $\mathbb{E}_{D|x_{1:n}}$ takes the expectation *w.r.t.* $p(D|x_{1:n})$. In this section, we derive a tractable approximation of this predictive distribution based on a detailed analysis of the posterior.

## 4.1  Posterior Analysis

Let $D \sim \mathrm{DP}(\alpha\mu)$ and $\theta_1, \ldots, \theta_n$ be iid samples from $D$, $\{C_1, \ldots, C_K\}$ be a partition of $\{1, \ldots, n\}$ such that $\theta_i$ for all $i \in C_k$ are identical, and $\phi_k = \theta_i \; \forall i \in C_k$. Then the posterior distribution of $D$ remains a DP, as $D|\theta_{1:n} \sim \mathrm{DP}(\tilde{\alpha}\tilde{\mu})$, where $\tilde{\alpha} = \alpha + n$, and

$$\tilde{\mu} = \frac{\alpha}{\alpha+n}\mu + \sum_{k=1}^{K} \frac{|C_k|}{\alpha+n} \delta_{\phi_k}. \tag{6}$$

The atoms are generally unobservable, and therefore it is more interesting in practice to consider the posterior distribution of $D$ given the observed samples. For this purpose, we derive the lemma below that provides a constructive characterization of the posterior distribution given both the observed samples $x_{1:n}$ and the partition $z$.

**Lemma 1.** *Consider the DPMM in Eq.(3). Drawing a sample from the posterior distribution $p(D|z_{1:n}, x_{1:n})$ is equivalent to constructing a random probability measure as follows*

$$\beta_0 D' + \sum_{k=1}^{K} \beta_k \delta_{\phi_k},$$

$$\text{with } D' \sim \mathrm{DP}(\alpha\mu), \; (\beta_0, \beta_1, \ldots, \beta_k) \sim \mathrm{Dir}(\alpha, m_1, \ldots, m_K), \; \phi_k \sim \mu|_{C_k}. \tag{7}$$

*Here, $m_k = |C_k|$, $\mu|_{C_k}$ is a posterior distribution given by i.e. $\mu|_{C_k}(d\theta) \propto \mu(d\theta) \prod_{i \in C_k} F(x_i|\theta)$.*

This lemma immediately follows from the Theorem 2 in [10] as DP is a special case of the so-called *Normalized Random Measures with Independent Increments (NRMI)*. It is worth emphasizing that $p(D|x, z)$ is no longer a Dirichlet process, as the locations of the atoms $\phi_1, \ldots, \phi_K$ are non-deterministic, instead they follow the posterior distributions $\mu|_{C_k}$.

By marginalizing out the partition $z_{1:n}$, we obtain the posterior distribution $p(D|x_{1:n})$:

$$p(D|x_{1:n}) = \sum_{z_{1:n}} p(z_{1:n}|x_{1:n}) p(D|x_{1:n}, z_{1:n}). \tag{8}$$

Let $\{C_1^{(z)}, \ldots, C_K^{(z)}\}$ be the partition corresponding to $z_{1:n}$, we have

$$p(z_{1:n}|x_{1:n}) \propto p_{CRF}(z_{1:n}|\alpha) \prod_{k=1}^{K^{(z)}} \int \mu(d\phi_k) \prod_{i \in C_k^{(z)}} F(x_i|\phi_k). \tag{9}$$

## 4.2 Variational Approximation

Computing the predictive distribution based on Eq.(8) requires enumerating all possible partitions, which grow exponentially as $n$ increases. To tackle this difficulty, we resort to *variational approximation*, that is, to choose a *tractable* distribution to approximate $p(D|x_{1:n}, z_{1:n})$.

In particular, we consider a family of random probability measures that can be expressed as follows:

$$q(D|\rho, \nu) = \sum_{z_{1:n}} \prod_{i=1}^{n} \rho_i(z_i) q_\nu^{(z)}(D|z_{1:n}). \qquad (10)$$

Here, $q_\nu^{(z)}(D|z_{1:n})$ is a stochastic process conditioned on $z_{1:n}$, defined as

$$q_\nu^{(z)}(D|z_{1:n}) \overset{d}{\sim} \beta_0 D' + \sum_{k=1}^{K} \beta_k \delta_{\phi_k},$$

$$\text{with } D' \sim \mathrm{DP}(\alpha\mu), \ (\beta_0, \beta_1, \ldots, \beta_K) \sim \mathrm{Dir}(\alpha, m_1^{(z)}, \ldots, m_K^{(z)}), \ \phi_k \sim \nu_k. \quad (11)$$

Here, we use $\overset{d}{\sim}$ to indicate that drawing a sample from $q_\nu^{(z)}$ is equivalent to constructing one according to the right hand side. In addition, $m_k^{(z)} = |C_k^{(z)}|$ is the cardinality of the $k$-th cluster *w.r.t.* $z_{1:n}$, and $\nu_k$ is a distribution over component parameters that is independent from $z$.

The variational construction in Eq.(10) and (11) is similar to Eq.(7) and (8), except for two significant differences: (1) $p(z_{1:n}|x_{1:n})$ is replaced by a product distribution $\prod_i \rho_i(z_i)$, and (2) $\mu|_{C_k}$, which depends on $z_{1:n}$, is replaced by an independent distribution $\nu_k$. With this design, $z_i$ for different $i$ and $\phi_k$ for different $k$ are independent *w.r.t.* $q$, thus resulting in a tractable predictive law below: Let $q$ be a random probability measure given by Eq.(10) and (11), then

$$\mathbb{E}_{q(D|\rho,\nu)}\left[p(\theta'|D)\right] = \frac{\alpha}{\alpha + n}\mu(\theta') + \sum_{k=1}^{K} \frac{\sum_{i=1}^{n} \rho_i(k)}{\alpha + n}\nu_k(\theta'). \qquad (12)$$

The approximate posterior has two sets of parameters: $\rho \triangleq (\rho_1, \ldots, \rho_n)$ and $\nu \triangleq (\nu_1, \ldots, \nu_n)$. With this approximation, the task of Bayesian learning reduces to the problem of finding the optimal setting of these parameters such that $q(D|\rho, \nu)$ best approximates the true posterior distribution.

## 4.3 Sequential Approximation

The first problem here is to determine the value of $K$. A straightforward approach is to fix $K$ to a large number as in the truncated methods. This way, however, would incur substantial computational costs on unnecessary components. We take a different approach here. Rather than randomly initializing a fixed number of components, we begin with an empty model (*i.e.* $K = 1$) and progressively refine the model as samples come in, adding new components on the fly when needed.

Specifically, when the first sample $x_1$ is observed, we introduce the first component and denote the posterior for this component by $\nu_1$. As there is only one component at this point, we have $z_1 = 1$, *i.e.* $\rho_1(z_1 = 1) = 1$, and the posterior distribution over the component parameter is $\nu_1^{(1)}(d\theta) \propto \mu(d\theta)F(x_1|\theta)$. Samples are brought in sequentially. In particular, we compute $\rho_i$, and update $\nu^{(i-1)}$ to $\nu^i$ upon the arrival of the $i$-th sample $x_i$.

Suppose we have $\rho = (\rho_1, \ldots, \rho_i)$ and $\nu^{(i)} = (\nu_1^{(i)}, \ldots, \nu_K^{(i)})$ after processing $i$ samples. To explain $x_{i+1}$, we can use either of the $K$ existing components or introduce a new component $\phi_{k+1}$. Then the posterior distribution of $z_{i+1}, \phi_1, \ldots, \phi_{K+1}$ given $x_1, \ldots, x_n, x_{n+1}$ is

$$p(z_{i+1}, \phi_{1:K+1}|x_{1:i+1}) \propto p(z_{i+1}, \phi_{1:K+1}|x_{1:i})p(x_{i+1}|z_{i+1}, \phi_{1:K+1}). \qquad (13)$$

Using the tractable distribution $q(\cdot|\rho_{1:i}, \nu^{(i)})$ in Eq.(10) to approximate the posterior $p(\cdot|x_{1:i})$, we get

$$p(z_{i+1}, \phi_{1:K+1}|x_{1:i+1}) \propto q(z_{i+1}|\rho_{1:i}, \nu^{(i)})p(x_{i+1}|z_{i+1}, \phi_{1:K+1}). \qquad (14)$$

Then, the optimal settings of $q_{i+1}$ and $\nu^{(i+1)}$ that minimizes the Kullback-Leibler divergence between $q(z_{i+1}, \phi_{1:K+1}|q_{1:i+1}, \nu^{(i+1)})$ and the approximate posterior in Eq.(14) are given as follows:

$$\rho_{i+1} \propto \begin{cases} w_k^{(i)} \int_\theta F(x_{i+1}|\theta)\nu_k^{(i)}(d\theta) & (k \le K), \\ \alpha \int_\theta F(x_{i+1}|\theta)\mu(d\theta) & (k = K+1), \end{cases} \qquad (15)$$

---

**Algorithm 1** Sequential Bayesian Learning of DPMM (for conjugate cases).

---

**Require:** base measure params: $\lambda, \lambda_0$, observed samples: $x_1, \ldots, x_n$, and threshold $\epsilon$

    Let $K = 1$, $\rho_1(1) = 1$, $w_1 = \rho_1$, $\zeta_1 = \phi(x_1)$, and $\zeta_1' = 1$.

    **for** $i = 2 : n$ **do**

        $T_i \leftarrow T(x_i)$, and $b_i \leftarrow b(x_i)$

        marginal log-likelihood: $h_i(k) \leftarrow \begin{cases} B(\zeta_k + T_i, \zeta_k' + \tau) - B(\zeta_k, \zeta_k') - b_i & (k = 1, \ldots, K) \\ B(\lambda + T_i, \lambda' + \tau) - B(\lambda, \lambda') - b_i & (k = K + 1) \end{cases}$

        $\rho_i(k) \leftarrow w_k e^{h_i(k)} / \sum_l w_l e^{h_i(l)}$ for $k = 1, \ldots, K + 1$ with $w_{K+1} = \alpha$

        **if** $\rho_i(K + 1) > \epsilon$ **then**

            $w_k \leftarrow w_k + \rho_i(k)$, $\zeta_k \leftarrow \zeta_k + \rho_i(k)T_i$, and $\zeta_k' \leftarrow \zeta_k' + \rho_i(k)\tau$, for $k = 1, \ldots, K$

            $w_{K+1} \leftarrow \rho_i(K + 1)$, $\zeta_{K+1} \leftarrow \rho_i(K + 1)T_i$, and $\zeta_{K+1}' \leftarrow \rho_i(K + 1)\tau$

            $K \leftarrow K + 1$

        **else**

            re-normalize $\rho_i$ such that $\sum_{k=1}^{K} \rho_i(k) = 1$

            $w_k \leftarrow w_k + \rho_i(k)$, $\zeta_k \leftarrow \zeta_k + \rho_i(k)T_i$, and $\zeta_k' \leftarrow \zeta_k' + \rho_i(k)\tau$, for $k = 1, \ldots, K$

        **end if**

    **end for**

---

with $w_k^{(i)} = \sum_{j=1}^{i} \rho_j(k)$, and

$$\nu_k^{(i+1)}(d\theta) \propto \begin{cases} \mu(d\theta) \prod_{j=1}^{i+1} F(x_j|\theta)^{\rho_j(k)} & (k \le K), \\ \mu(d\theta) F(x_{i+1}|\theta)^{\rho_{i+1}(k)} & (k = K + 1). \end{cases} \qquad (16)$$

**Discussion.** There is a key distinction between this approximation scheme and conventional approaches: Instead of seeking the approximation of $p(D|x_{1:n})$, which is very difficult ($D$ is infinite) and unnecessary (only a finite number of components are useful), we try to approximate the posterior of a finite subset of latent variables that are truly relevant for prediction, namely $z$ and $\phi_{1:K+1}$.

This sequential approximation scheme introduces a new component for each sample, resulting in $n$ components over the entire dataset. This, however, is unnecessary. We find empirically that for most samples, $\rho_i(K + 1)$ is negligible, indicating that the sample is adequately explained by existing component, and there is no need of new components. In practice, we set a small value $\epsilon$ and increase $K$ only when $\rho_i(K + 1) > \epsilon$. This simple strategy is very effective in controlling the model size.

## 5 Algorithm and Implementation

This section discusses the implementation of the sequential Bayesian learning algorithm under two different circumstances: (1) $\mu$ and $F$ are exponential family distributions that form a conjugate pair, and (2) $\mu$ is not a conjugate prior *w.r.t.* $F$.

**Conjugate Case.** In general, when $\mu$ is conjugate to $F$, they can be written as follows:

$$\mu(d\theta|\lambda, \lambda') = \exp\left(\lambda^T \eta(\theta) - \lambda' A(\theta) - B(\lambda, \lambda')\right) h(d\theta), \qquad (17)$$

$$F(x|\theta) = \exp\left(\eta(\theta)^T T(x) - \tau A(\theta) - b(x)\right). \qquad (18)$$

Here, the prior measure $\mu$ has a pair of natural parameters: $(\lambda, \lambda')$. Conditioned on a set of observations $x_1, \ldots, x_n$, the posterior distribution remains in the same family as $\mu$ with parameters $(\lambda + \sum_{i=1}^{n} T(x_i), \lambda' + n\tau)$. In addition, the marginal likelihood is given by

$$\int_\theta F(x|\theta)\mu(d\theta|\lambda, \lambda') = \exp\left(B(\lambda + T(x), \lambda' + \tau) - B(\lambda, \lambda') - b(x)\right). \qquad (19)$$

In such cases, both the base measure $\mu$ and the component-specific posterior measures $\nu_k$ can be represented using the natural parameter pairs, which we denote by $(\lambda, \lambda')$ and $(\zeta_k, \zeta_k')$. With this notation, we derive a sequential learning algorithm for conjugate cases, as shown in Alg 1.

**Non-conjugate Case.** In practical models, it is not uncommon that $\mu$ and $F$ are not a conjugate pair. Unlike in the conjugate cases discussed above, there exist no formulas to update posterior

parameters or to compute marginal likelihood in general. Here, we propose to address this issue using stochastic optimization. Consider a posterior distribution given by $p(\theta|x_{1:n}) \propto \mu(\theta)\prod_{i=1}^{n} F(x_i|\theta)$. A stochastic optimization method finds the MAP estimate of $\theta$ through update steps as below:

$$\theta \leftarrow \theta + \sigma_i \left(\nabla_\theta \log \mu(\theta) + n\nabla_\theta \log F(x_i|\theta)\right). \tag{20}$$

The basic idea here is to use the gradient computed at a particular sample $x_i$ to approximate the true gradient. This procedure converges to a (local) maximum, as long as the step size $\sigma_i$ satisfy $\sum_{i=1}^{\infty} \sigma_i = \infty$ and $\sum_{i=1}^{\infty} \sigma_i^2 < \infty$.

Incorporating the stochastic optimization method into our algorithm, we obtain a variant of Alg 1. The general procedure is similar, except for the following changes: (1) It maintains point estimates of the component parameters instead of the posterior, which we denote by $\hat{\phi}_1, \ldots, \hat{\phi}_K$. (2) It computes the log-likelihood as $h_i(k) = \log F(x_i|\hat{\phi}_k)$. (3) The estimates of the component parameters are updated using the formula below:

$$\hat{\phi}_k^{(i)} \leftarrow \hat{\phi}_k^{(i-1)} + \sigma_i \left(\nabla_\theta \log \mu(\theta) + n\rho_i(k)\nabla_\theta \log F(x_i|\theta)\right). \tag{21}$$

Following the common practice of stochastic optimization, we set $\sigma_i = i^{-\kappa}/n$ with $\kappa \in (0.5, 1]$.

**Prune and Merge.** As opposed to random initialization, components created during this sequential construction are often truly needed, as the decisions of creating new components are based on knowledge accumulated from previous samples. However, it is still possible that some components introduced at early iterations would become less useful and that multiple components may be similar. We thus introduce a mechanism to remove undesirable components and merge similar ones.

We identify opportunities to make such adjustments by looking at the weights. Let $\tilde{w}_k^{(i)} = w_k^{(i)}/\sum_l w_l^{(i)}$ (with $w_k^{(i)} = \sum_{j=1}^{i} \rho_j(k)$) be the relative weight of a component at the $i$-th iteration. Once the relative weight of a component drops below a small threshold $\varepsilon_r$, we remove it to save unnecessary computation on this component in the future.

The similarity between two components $\phi_k$ and $\phi_{k'}$ can be measured in terms of the distance between $\rho_i(k)$ and $\rho_i(k')$ over all processed samples, as $d_\rho(k, k') = i^{-1}\sum_{j=1}^{i}|\rho_j(k) - \rho_j(k')|$. We increment $\rho_i(k)$ to $\rho_i(k) + \rho_i(k')$ when $\phi_k$ and $\phi_k'$ are merged (i.e. $d_\rho(k, k') < \varepsilon_d$). We also merge the associated sufficient statistics (for conjugate case) or take an weighted average of the parameters (for non-conjugate case). Generally, there is no need to perform such checks at every iteration. Since computing this distance between a pair of components takes $O(n)$, we propose to examine similarities at an $O(i \cdot K)$-interval so that the amortized complexity is maintained at $O(nK)$.

**Discussion.** As compared to existing methods, the proposed method has several important advantages. First, it builds up the model on the fly, thus avoiding the need of randomly initializing a set of components as required by truncation-based methods. The model learned in this way can be readily extended (*e.g.* adding more components or adapting existing components) when new data is available. More importantly, the algorithm can learn the model in one pass, without the need of iterative updates over the data set. This distinguishes it from MCMC methods and conventional variational learning algorithms, making it a great fit for large scale problems.

# 6 Experiments

To test the proposed algorithm, we conducted experiments on both synthetic data and real world applications – modeling image patches and document clustering. All algorithms are implemented using Julia [2], a new language for high performance technical computing.

## 6.1 Synthetic Data

First, we study the behavior of the proposed algorithm on synthetic data. Specifically, we constructed a data set comprised of 10000 samples in 9 Gaussian clusters of unit variance. The distances between these clusters were chosen such that there exists moderate overlap between neighboring clusters. The estimation of these Gaussian components are based on the DPMM below:

$$D \sim \text{DP}\left(\alpha \cdot \mathcal{N}(\mathbf{0}, \sigma_p^2 \mathbf{I})\right), \quad \boldsymbol{\theta}_i \sim D, \quad \mathbf{x}_i \sim \mathcal{N}(\boldsymbol{\theta}_i, \sigma_x^2 \mathbf{I}). \tag{22}$$

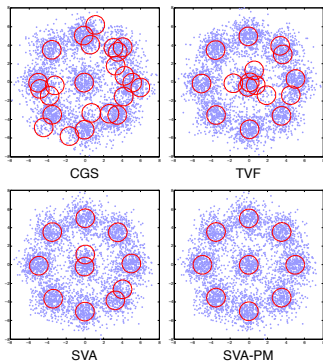

Figure 1: Gaussian clusters on synthetic data obtained using different methods. Both MC-SM and SVA-PM identified the 9 clusters correctly. The result of MC-SM is omitted here, as it looks the same as SVA-PM.

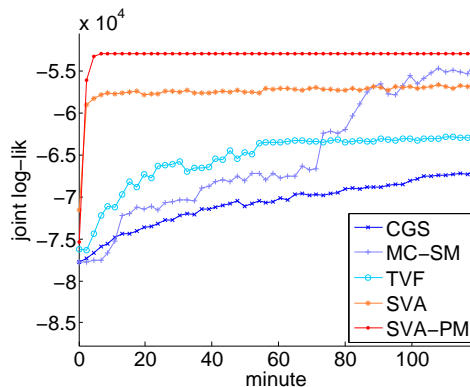

Figure 2: Joint log-likelihood on synthetic data as functions of run-time. The likelihood values were evaluated on a held-out testing set. (Best to view with color)

Here, we set $\alpha = 1$, $\sigma_p = 100$ and $\sigma_x = 1$.

We tested the following inference algorithms: *Collapsed Gibbs sampling (CGS)* [12], *MCMC with Split-Merge (MC-SM)* [6], *Truncation-Free Variational Inference (TFV)* [21], *Sequential Variational Approximation (SVA)*, and its variant *Sequential Variational Approximation with Prune and Merge (SVA-PM)*. For CGS, MC-SM, and TFV, we run the updating procedures iteratively for one hour, while for SVA and SVA-PM, we run only one-pass.

Figure 1 shows the resulting components. CGS and TFV yield obviously redundant components. This corroborates observations in previous work [9]. Such nuisances are significantly reduced in SVA, which only occasionally brings in redundant components. The key difference that leads to this improvement is that CGS and TFV rely on random initialization to bootstrap the algorithm, which would inevitably introduce similar components, while SVA leverages information gained from previous samples to decide whether new components are needed. Both MC-SM and SVA-PM produce desired mixtures, demonstrating the importance of an explicit mechanism to remove redundancy.

Figure 2 plots the traces of joint log-likelihoods evaluated on a held-out set of samples. We can see that SVA-PM quickly reaches the optimal solution in a matter of seconds. SVA also gets to a reasonable solution within seconds, and then the progress slows down. Without the prune-and-merge steps, it takes much longer for redundant components to fade out. MC-SM eventually reaches the optimal solution after many iterations. Methods relying on local updates, including CGS and TFV, did not even come close to the optimal solution within one hour. These results clearly demonstrate that our progressive strategy, which gradually constructs the model through a series of informed decisions, is much more efficient than random initialization followed by iterative updating.

## 6.2 Modeling Image Patches

Image patches, which capture local characteristics of images, play a fundamental role in various computer vision tasks, such as image recovery and scene understanding. Many vision algorithms rely on a patch dictionary to work. It has been a common practice in computer vision to use parametric methods (*e.g.* K-means) to learn a dictionary of fixed size. This approach is inefficient when large datasets are used. It is also difficult to be extended when new data with a fixed $K$.

To tackle this problem, we applied our method to learn a nonparametric dictionary from the SUN database [24], a large dataset comprised of over $130K$ images, which capture a broad variety of scenes. We divided all images into two disjoint sets: a training set with $120K$ images and a testing set with $10K$. We extracted 2000 patches of size $32 \times 32$ from each image, and characterize each patch by a 128-dimensional SIFT feature. In total, the training set contains $240M$ feature vectors. We respectively run TFV, SVA, and SVA-SM to learn a DPMM from the training set, based on the

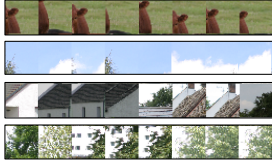

Figure 3: Examples of image patche clusters learned using SVA-PM. Each row corresponds to a cluster. We can see similar patches are in the same cluster.

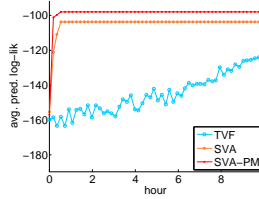

Figure 4: Average log-likelihood on image modeling as functions of run-time.

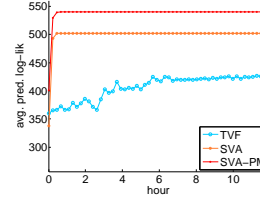

Figure 5: Average log-likelihood of document clusters as functions of run-time.

formulation given in Eq.(22), and evaluate the average predictive log-likelihood over the testing set as the measure of performance. Figure 3 shows a small subset of patch clusters obtained using SVA-PM.

Figure 4 compares the trajectories of the average log-likelihoods obtained using different algorithms. TFV takes multiple iterations to move from a random configuration to a sub-optimal one and get trapped in a local optima. SVA steadily improves the predictive performance as it sees more samples. We notice in our experiments that even without an explicit redundancy-removal mechanism, some unnecessary components can still get removed when their relative weights decreases and becomes negligible. SVM-PM accelerates this process by explicitly merging similar components.

## 6.3 Document Clustering

Next, we apply the proposed method to explore categories of documents. Unlike standard topic modeling task, this is a higher level application that builds on top of the topic representation. Specifically, we first obtain a collection of $m$ topics from a subset of documents, and characterize all documents by topic proportions. We assume that the topic proportion vector is generated from a category-specific Dirichlet distribution, as follows

$$D \sim \text{DP}\left(\alpha \cdot \text{Dir}_{sym}(\gamma_p)\right), \quad \boldsymbol{\theta}_i \sim D, \quad \mathbf{x}_i \sim \text{Dir}(\gamma_x \boldsymbol{\theta}_i). \tag{23}$$

Here, the base measure is a symmetric Dirichlet distribution. To generate a document, we draw a mean probability vector $\boldsymbol{\theta}_i$ from $D$, and generates the topic proportion vector $\mathbf{x}_i$ from $\text{Dir}(\gamma_x \boldsymbol{\theta}_i)$. The parameter $\gamma_x$ is a design parameter that controls how far $\mathbf{x}_i$ may deviate from the category-specific center $\boldsymbol{\theta}_i$. Note that this is not a conjugate model, and we use stochastic optimization instead of Bayesian updates in SVA (see section 5).

We performed the experiments on the New York Times database, which contains about $1.8M$ articles from year 1987 to 2007. We pruned the vocabulary to 5000 words by removing stop words and those with low TF-IDF scores, and obtained 150 topics by running LDA [3] on a subset of $20K$ documents. Then, each document is represented by a 150-dimensional vector of topic proportions. We held out $10K$ documents for testing and use the remaining to train the DPMM. We compared SVA, SVA-PM, and TVF. The traces of log-likelihood values are shown in Figure 5. We observe similar trends as above: SVA and SVA-PM attains better solution more quickly, while TVF is less efficient and is prune to being trapped in local maxima. Also, TVF tends to generate more components than necessary, while SVA-PM maintains a better performance using much less components.

## 7 Conclusion

We presented an online Bayesian learning algorithm to estimate DP mixture models. The proposed method does not require random initialization. Instead, it can reliably and efficiently learn a DPMM from scratch through sequential approximation in a single pass. The algorithm takes in data in a streaming fashion, and thus can be easily adapted to new data. Experiments on both synthetic data and real applications have demonstrated that our algorithm achieves remarkable speedup – it can attain nearly optimal configuration within seconds or minutes, while mainstream methods may take hours or even longer. It is worth noting that the approximation is derived based on the predictive law of DPMM. It is an interesting future direction to investigate how it can be generalized to a broader family of BNP models, such as HDP, Pitman-Yor processes, and NRMIs [10].

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
