[Supplementary Material · psb_supp.pdf]

# Supplemental Document

**Dahua Lin**
Toyota Technological Institute at Chicago
`dhlin@ttic.edu`

## Abstract

This document provides proofs of theorems presented in the paper, as well as computational details such as the formulas to update components.

## 1 Component Update

This section presents technical details about component updates.

### 1.1 Gaussian Models with Fixed Covariance

Consider the following formulation:

$$\boldsymbol{\theta} \sim \mathcal{N}(\boldsymbol{\mu}_p, \boldsymbol{\Sigma}_p), \quad \text{and} \quad \mathbf{x}_i | \boldsymbol{\theta} \sim \mathcal{N}(\boldsymbol{\theta}, \boldsymbol{\Sigma}_x), \ \forall i = 1, \ldots, n. \tag{1}$$

Here, each sample $\mathbf{x}$ is generated from a $d$-dimensional Gaussian distribution with mean $\boldsymbol{\theta}$ and a fixed covariance $\boldsymbol{\Sigma}_x$, and the $\boldsymbol{\theta}$ itself is generated from a prior Gaussian with mean $\boldsymbol{\mu}$ and covariance $\boldsymbol{\Sigma}_p$. We can rewrite these distributions in the form of exponential family distributions as follows:

$$p(\boldsymbol{\theta}) = \exp\left( -\frac{1}{2}\boldsymbol{\theta}^T\boldsymbol{\Sigma}_p^{-1}\boldsymbol{\theta} + \boldsymbol{\mu}_p^T\boldsymbol{\Sigma}_p^{-1}\boldsymbol{\theta} - \frac{1}{2}\boldsymbol{\mu}_p^T\boldsymbol{\Sigma}_p^{-1}\boldsymbol{\mu}_p - \frac{1}{2}\left( d\log(2\pi) + \log|\boldsymbol{\Sigma}_p| \right) \right), \tag{2}$$

$$F(\mathbf{x}|\boldsymbol{\theta}) = \exp\left( -\frac{1}{2}\boldsymbol{\theta}^T\boldsymbol{\Sigma}_x^{-1}\boldsymbol{\theta} + \boldsymbol{\theta}^T\boldsymbol{\Sigma}_x^{-1}\mathbf{x} - \frac{1}{2}\mathbf{x}^T\boldsymbol{\Sigma}_x^{-1}\mathbf{x} - \frac{1}{2}\left( d\log(2\pi) + \log|\boldsymbol{\Sigma}_x| \right) \right). \tag{3}$$

Let $\mathbf{h}_p = \boldsymbol{\Sigma}_p^{-1}\boldsymbol{\mu}_p$, $mJ_p = \boldsymbol{\Sigma}_p^{-1}$, and

$$B(\mathbf{h}_p, \boldsymbol{J}_p) = \frac{1}{2}\left( \mathbf{h}_p^T\boldsymbol{J}_p^{-1}\mathbf{h}_p + d\log(2\pi) - \log|\boldsymbol{J}_p| \right) = \frac{1}{2}\left( \boldsymbol{\mu}_p^T\boldsymbol{\Sigma}_p^{-1}\boldsymbol{\mu}_p + d\log(2\pi) + \log|\boldsymbol{\Sigma}_p| \right).$$

Then Eq.(2) can be rewritten as

$$p(\theta|\mathbf{h}, \boldsymbol{J}) = \exp\left( \mathbf{h}^T\boldsymbol{\theta} + \langle \boldsymbol{J}_p, -\frac{1}{2}\boldsymbol{\theta}\boldsymbol{\theta}^T \rangle - B(\mathbf{h}_p, \boldsymbol{J}_p) \right). \tag{4}$$

In addition, let $\boldsymbol{J}_x = \boldsymbol{\Sigma}_x^{-1}$, $T(\mathbf{x}) = \boldsymbol{\Sigma}_x^{-1}\mathbf{x} = \boldsymbol{J}_x\mathbf{x}$, $A(\boldsymbol{\theta}) = -\frac{1}{2}\boldsymbol{\theta}\boldsymbol{\theta}^T$, and

$$b(\mathbf{x}) = \frac{1}{2}\left( \mathbf{x}^T\boldsymbol{\Sigma}_x^{-1}\mathbf{x} + d\log(2\pi) + \log|\boldsymbol{\Sigma}_x| \right).$$

We then rewrite Eq.(3) as

$$F(\mathbf{x}|\boldsymbol{\theta}) = \exp\left( \boldsymbol{\theta}^T T(\mathbf{x}) + \langle \boldsymbol{J}_x, A(\boldsymbol{\theta}) \rangle - b(\mathbf{x}) \right)$$
$$= \exp\left( \boldsymbol{\theta}^T\left( \boldsymbol{\Sigma}_x^{-1}\mathbf{x} \right) + \langle \boldsymbol{J}_x, -\frac{1}{2}\boldsymbol{\theta}\boldsymbol{\theta}^T \rangle - b(\mathbf{x}) \right). \tag{5}$$

It is not difficult to see from the analysis above that, conditioned on $\mathbf{x}$, the posterior distribution of $\boldsymbol{\theta}$ remains a Gaussian, whose canonical parameters are updated from $(\mathbf{h}_p, \boldsymbol{J}_p)$ to $(\mathbf{h}_p + \boldsymbol{J}_x \mathbf{x}, \boldsymbol{J}_p + \boldsymbol{J}_x)$.

The log-marginal likelihood of $\mathbf{x}$ is thus given by

$$
\begin{aligned}
\log p(\mathbf{x}|\mathbf{h}_p, \boldsymbol{J}_p) &= \log \int_{\boldsymbol{\theta}} F(\mathbf{x}|\boldsymbol{\theta}) p(\boldsymbol{\theta}|\mathbf{h}, \boldsymbol{J}) d\theta \\
&= B(\mathbf{h}_p + \boldsymbol{J}_x \mathbf{x}, \boldsymbol{J}_p + \boldsymbol{J}_x) - B(\mathbf{h}_p, \boldsymbol{J}_p) - b(\mathbf{x}) \\
&= \frac{1}{2}(t_1 + t_2),
\end{aligned}
\tag{6}
$$

with

$$
t_1 = (\boldsymbol{\Sigma}_p^{-1}\boldsymbol{\mu}_p + \boldsymbol{\Sigma}_x^{-1}\mathbf{x})^T (\boldsymbol{\Sigma}_p^{-1} + \boldsymbol{\Sigma}_x^{-1})^{-1} (\boldsymbol{\Sigma}_p^{-1}\boldsymbol{\mu}_p + \boldsymbol{\Sigma}_x^{-1}\mathbf{x}) - \boldsymbol{\mu}_p^T \boldsymbol{\Sigma}_p^{-1}\boldsymbol{\mu}_p - \mathbf{x}^T \boldsymbol{\Sigma}_x^{-1}\mathbf{x}, \tag{7}
$$

$$
t_2 = \left(d\log(2\pi) - \log|\boldsymbol{\Sigma}_x^{-1} + \boldsymbol{\Sigma}_p^{-1}|\right) - (d\log(2\pi) + \log|\boldsymbol{\Sigma}_p|) - (d\log(2\pi) + \log|\boldsymbol{\Sigma}_x|). \tag{8}
$$

Particularly, these formulas can be further simplified as below:

$$
t_1 = -(\mathbf{x} - \boldsymbol{\mu})^T (\boldsymbol{\Sigma}_x + \boldsymbol{\Sigma}_p)^{-1} (\mathbf{x} - \boldsymbol{\mu}), \tag{9}
$$

and

$$
\begin{aligned}
t_2 &= -d\log(2\pi) - \left(\log|\boldsymbol{\Sigma}_x^{-1} + \boldsymbol{\Sigma}_p^{-1}| + \log|\boldsymbol{\Sigma}_p| + \log|\boldsymbol{\Sigma}_x|\right) \\
&= -d\log(2\pi) - \log|\boldsymbol{\Sigma}_x(\boldsymbol{\Sigma}_x^{-1} + \boldsymbol{\Sigma}_p^{-1})\boldsymbol{\Sigma}_p| \\
&= -d\log(2\pi) - \log|\boldsymbol{\Sigma}_x + \boldsymbol{\Sigma}_p|.
\end{aligned}
\tag{10}
$$

Therefore, we have

$$
\log p(\mathbf{x}|\mathbf{h}_p, \boldsymbol{J}_p) = -\frac{1}{2}\left((\mathbf{x} - \boldsymbol{\mu}_p)^T (\boldsymbol{\Sigma}_x + \boldsymbol{\Sigma}_p)^{-1}(\mathbf{x} - \boldsymbol{\mu}_p) + d\log(2\pi) + \log|\boldsymbol{\Sigma}_x + \boldsymbol{\Sigma}_p|\right). \tag{11}
$$

This is consistent with the fact with $\mathbf{x} \sim \mathcal{N}(\boldsymbol{\mu}, \boldsymbol{\Sigma}_p + \boldsymbol{\Sigma}_x)$ when $\boldsymbol{\theta}$ is marginalized out.

In particular, when $\boldsymbol{\Sigma}_p = \sigma_p^2 \boldsymbol{I}$ and $\boldsymbol{\Sigma}_x = \sigma_x^2 \boldsymbol{I}$, we have (with $\mathbf{h}_p = \sigma_p^{-2}\boldsymbol{\mu}_p$):

$$
p(\boldsymbol{\theta}|\mathbf{h}_p, \sigma_p^{-2}\boldsymbol{I}) = \exp\left(\mathbf{h}_p^T \boldsymbol{\theta} - \frac{1}{2}\sigma_p^{-2}||\boldsymbol{\theta}||^2 - B(\mathbf{h}_p, \sigma_p^{-2}\boldsymbol{I})\right), \tag{12}
$$

$$
F(\boldsymbol{\theta}|\boldsymbol{\theta}, \sigma_x^{-2}\boldsymbol{I}) = \exp\left(\boldsymbol{\theta}^T (\sigma_x^{-2}\mathbf{x}) - \frac{1}{2}\sigma_x^{-2}||\boldsymbol{\theta}||^2 - b(\mathbf{x})\right). \tag{13}
$$

Here,

$$
B(\mathbf{h}_p, \sigma_p^{-2}\boldsymbol{I}) = \frac{1}{2}\left(\sigma_p^2||\mathbf{h}_p||^2 + d\log(2\pi\sigma_p^2)\right), \quad \text{and} \quad b(\mathbf{x}) = \frac{1}{2}\left(\sigma_x^{-2}||\mathbf{x}||^2 + d\log(2\pi\sigma_x^2)\right). \tag{14}
$$

Update of posterior parameter is given by the formulas below:

$$
\mathbf{h} \leftarrow \mathbf{h}_p + \sigma_x^{-2}\mathbf{x}, \quad \text{and} \quad \sigma^{-2} \leftarrow \sigma_p^{-2} + \sigma_x^{-2}. \tag{15}
$$