[Reviews · NeurIPS 2013]

Submitted by Assigned_Reviewer_1

This paper proposed a sequential Bayesian inference of the Dirichlet process mixture models.
The proposed algorithm is simple, but outperformed the existing work such as truncation-free variational inference.

My concern is that the model selection depends on \rho(K+1)<\epsilon.
That is, the performance of this method depends on \epsilon.
The effective value of \epsilon seems to be unstable to the property of data and the type of a mixture component.
How do you decide \epsilon and how robust is it?

The highly related work is
DAVID J. NOTT1, XIAOLE ZHANG2, CHRISTOPHER YAU2 & AJAY JASRA1
A sequential algorithm for fast fi tting of Dirichlet process mixture
models
http://www.stat.nus.edu.sg/~staja/vsugs2.pdf

I'd like you to comment on this paper.
Summary: This paper proposed a sequential Bayesian inference of the Dirichlet process mixture models.
The proposed algorithm is simple, but outperformed a more complex method such as truncation-free variational inference.

Submitted by Assigned_Reviewer_4

In this paper the authors presented a sequential variational inference algorithm for Dirichlet process mixture models. The authors used a posterior characterization of normalized random measures with independent increments as the basis for a variational distribution that was then used on a sequential decomposition of the posterior. The algorithm was demonstrated on a Gaussian mixture model applied to real and synthetic data, and a non-conjugate DP mixture of Dirichlet distributions to cluster text documents.

This is a nice paper that aims to make a particular Bayesian nonparametric model useful to analyze massive data sets. The overall presentation is clear and well-motivated. The experiments are explained clearly and demonstrate advantages of the proposed algorithm.

I think that the impact of the paper in expanding the usefulness of Bayesian nonparametric models could be greatly improved by including a supplementary materials section with some proofs and derivations so that a wider audience can understand how the algorithm works and adapt it for their problem. Specifically, including the following would make the paper more self-contained and much clearer:
1. A proof of Lemma 1. The theorem that the proof is a result from is in a pretty dense paper. Restating the theorem and showing how Lemma 1 follows would be very helpful.
2. Derive Equation 12. The variational distribution is non-standard and it's pretty tedious to derive Equation 12.
3. Derive Equations 15 and 16 from Equation 14. The variational distribution is non-standard, the notation for the variational distribution changed from q(D) to q(z,phi), and the sequential variational framework is different enough from the standard setup that it would save some mathematical headaches for the reader to see how Equations 15 and 16 arise.

A few minor comments on the paper that should be addressed: This work should situate itself with respect to the work on sequential Monte Carlo algorithms for Dirichlet process mixture models which use the same sequential decomposition of the posterior. I think Equation 14 is missing the \phi's in the variational posterior. There are a few parameters of the algorithm, namely \epsilon_d and \epsilon_r, what were these set to for the experiments, or were they found with cross-validation? Lastly, the abbreviation CRF is not standard, why not CRP?
Summary: This is an interesting paper that introduces a sequential variational inference algorithm in order to make DP mixture models useful for massive data. The clarity and usefulness of the paper to the research community could be improved with a supplemental materials section with derivations of the results.

Submitted by Assigned_Reviewer_5

The authors present a fast algorithm for learning DPs using a sequential variational approximation to the predictive distribution. The experiments indicate significant speed-ups compared to state-of-the-art learning algorithms for DPs.

The paper is well-written and fairly easy to follow. The idea of sequential variational approximation to the predictive distribution seems novel (to the best of my knowledge) and the one-pass algorithm is attractive for applying DPs to large datasets. I feel the experimental section could have been better; other than that, I think this is a good paper.

Comments:

I believe the authors incorrectly use CRF (instead of CRP) to denote Chinese Restaurant process in (4) and elsewhere.

Line 358: The authors claim that the poor performance of CGS and TFV is due to random initialization. It might be a good idea to present results for CGS and TFV when they are initialized with the output of SVA to verify if this is indeed the case.

Experimental section:
- how sensitive are the results to the order in which the data is processed? perhaps the authors could present results for multiple shuffles of the dataset?
- What are the hyperparameter values in Section 6.2 and 6.3?
- How do CGS, MC-SM perform for the datasets in section 6.2 and section 6.3?
- comparison of K for different methods in section 6.2 and 6.3 might be helpful
- Section 6.3: The experimental setup is not clear. Is each document represented by a 150-dimensional feature vector? How are these features obtained for test documents?
- Some indication of sensitivity to \epsilon_r, \epsilon_d would be useful.

I encourage the authors to make their code publicly available.

Minor points:
\nu not defined in line 120
typo: "along is line" in line 74
line 150: "given by i.e."
Line 383: Not clear how there are 240M feature vectors. There should be only 2000 x 128 = 256K feature vectors per image right?

UPDATE (after rebuttal): The authors' response addressed my questions satisfactorily.
Summary: The idea of sequential variational approximation to the predictive distribution seems novel and appears to be a promising direction for scaling up Bayesian non-parametric models to large datasets. Experimental section could be improved.
Author Feedback

Author rebuttal: We thank all reviewers for the constructive feedback.

R1 asked about the setting of epsilon.

Empirically, we found that epsilon influences how fast new components are introduced. However, if this algorithm is applied to a large dataset, the final results (i.e. clustering of the samples) are not very sensitive to this value. To be more specific, if we set epsilon to a relatively higher value (i.e. being more conservative in adding new components), components are added in a slower pace, but each data cluster would eventually get its own component. On the other hand, setting this threshold too low would incur lots of false components and one has to resort to the prune-and-merge steps to clean up the model. We will provide a brief discussion about the influence of epsilon in the final version.

In our experiments, we conducted cross validation over the synthetic dataset and chose to set \epsilon to 0.001. This setting works pretty well across all the experiments presented in the paper. We plan to further investigate whether it is helpful to adjust its value adaptively on the fly.

R1 asked about comments on Nott et al’s paper.

Similar to our algorithm, the VSUGS method proposed in this paper takes a sequential updating approach, but relies on a different approximation. Particularly, what we approximate is a joint posterior over both data allocation and model parameters, while VSUGS is based on the approximating the posterior of data allocation. Also, VSUGS requires fixing a truncation level T in advance, which may lead to difficulties in practice (especially for large data). Our algorithm provides a way to tackle this, and no longer requires fixed truncation.

R2 suggested including supplementary materials with detailed proofs and derivations.

Whereas we have included a skeleton of the derivation in section 4, we do agree that providing details of the derivation in supplemental materials will definitely make this paper clearer and easier to follow. We appreciate this suggestion and will provide such materials along with our final version.

R3 suggested presenting results for CGS and TFV when they are initialized with the output of SVA (instead of random initialization).

We think this is a very good idea; as such an experiment will give a clearer idea about how different parts of the algorithm influence the final performance. We run a quick experiment to investigate this over the synthetic data, and found that using SVA for initialization did considerably improve the performance of CGS and TFV. However, these two methods still tend to introduce false or unnecessary components occasionally after initialization.

R3 have several questions regarding the experiment section:

(a) Sensitivity of the results to the order in which the data is processed: in our implementation of the experiments, the data set was randomly shuffled, and the algorithm performed similarly. However, there might exist (contrived) adverse order on which the algorithm may fail.
(b) Hyper-parameter values in section 6.2 & 6.3 are found via cross validation. Particularly, for section 6.2, we set alpha to 10.0 and sigma_x to 16.0; while for section 6.3, we set alpha to 10.0 and gamma_x to 2.0.
(c) Setup of section 6.3: it is worth noting that we are not trying to discover topics using a nonparametric model, instead, we simply run LDA to obtain the topics and use the topic proportion vector to represent each document. Then a DPMM model is used to cluster documents into groups.

R3 suggests making the codes publically available. We appreciate this suggestion, and we plan to release the codes together with the experiment scripts when the paper is published.

Finally, we thank both R2 and R3 for catching our mistake of using CRF in the place of CRP. We will correct this.